# Melatonin-Loaded Nanocarriers: New Horizons for Therapeutic Applications

**DOI:** 10.3390/molecules26123562

**Published:** 2021-06-10

**Authors:** Luiz Gustavo de Almeida Chuffa, Fábio Rodrigues Ferreira Seiva, Adriana Alonso Novais, Vinícius Augusto Simão, Virna Margarita Martín Giménez, Walter Manucha, Debora Aparecida Pires de Campos Zuccari, Russel J. Reiter

**Affiliations:** 1Department of Structural and Functional Biology, Institute of Biosciences, UNESP-São Paulo State University, Botucatu, São Paulo 18618-689, Brazil; luiz-gustavo.chuffa@unesp.br (L.G.d.A.C.); viniciusmorma@gmail.com (V.A.S.); 2Biological Science Center, Department of Biology, Luiz Meneghel Campus, Universidade Estadual do Norte do Paraná-UENP, Bandeirantes 86360-000, PR, Brazil; fabio.seiva@uenp.edu.br; 3Health Sciences Institute, Federal University of Mato Grosso, UFMT, Sinop 78607-059, MG, Brazil; aanovais@terra.com.br; 4Facultad de Ciencias Químicas y Tecnológicas, Instituto de Investigaciones en Ciencias Químicas, Universidad Católica de Cuyo, Sede San Juan 5400, Argentina; virchimg@hotmail.com; 5Laboratorio de Farmacología Experimental Básica y Traslacional. Área de Farmacología, Departamento de Patología, Facultad de Ciencias Médicas, Universidad Nacional de Cuyo, Mendoza 5500, Argentina; wmanucha@yahoo.com.ar; 6Instituto de Medicina y Biología Experimental de Cuyo, Consejo Nacional de Investigación Científica y Tecnológica (IMBECU-CONICET), Mendoza 5500, Argentina; 7Cancer Molecular Research Laboratory (LIMC), Department of Molecular Biology, FAMERP, São José do Rio Preto 15090-000, Brazil; zuccaridebora@gmail.com; 8Department of Cell Systems and Anatomy, UT Health, San Antonio, TX 78229, USA

**Keywords:** melatonin, nanocarriers, melatonin-loaded nanocarriers, nanoparticles, drug-delivery system

## Abstract

The use of nanosized particles has emerged to facilitate selective applications in medicine. Drug-delivery systems represent novel opportunities to provide stricter, focused, and fine-tuned therapy, enhancing the therapeutic efficacy of chemical agents at the molecular level while reducing their toxic effects. Melatonin (*N*-acetyl-5-methoxytriptamine) is a small indoleamine secreted essentially by the pineal gland during darkness, but also produced by most cells in a non-circadian manner from which it is not released into the blood. Although the therapeutic promise of melatonin is indisputable, aspects regarding optimal dosage, biotransformation and metabolism, route and time of administration, and targeted therapy remain to be examined for proper treatment results. Recently, prolonged release of melatonin has shown greater efficacy and safety when combined with a nanostructured formulation. This review summarizes the role of melatonin incorporated into different nanocarriers (e.g., lipid-based vesicles, polymeric vesicles, non-ionic surfactant-based vesicles, charge carriers in graphene, electro spun nanofibers, silica-based carriers, metallic and non-metallic nanocomposites) as drug delivery system platforms or multilevel determinations in various in vivo and in vitro experimental conditions. Melatonin incorporated into nanosized materials exhibits superior effectiveness in multiple diseases and pathological processes than does free melatonin; thus, such information has functional significance for clinical intervention.

## 1. Introduction

### 1.1. Tuning the Connection between Nanocarriers and Melatonin

Nanotechnology has been a well-known scientific field, since the 20th century when the Nobel laureate physicist Richard P. Feynman presented his lecture in 1959 entitled “There’s Plenty of Room at the Bottom” [1]. This revolutionary area is dedicated to the study, development, and manipulation of diverse biomaterials at the nanoscale level. In nanotechnology, nanocarriers (NCs) represent structures with lengths in two- or three-dimension orders > than 0.001 µm (1 nanometer) and smaller than 0.1 µm (100 nanometers) [2]. However, there are less strict definitions of nanoscale, which establish that nanostructures with sizes below 1000 nm may also be considered NCs [3]. Given the specific overall shape of the nanostructures, these formations can be found in 0D, 1D, 2D, or 3D [4]. Based on the fact that size and particular characteristics of these materials influence physicochemical properties of a substance (e.g., optical properties), nanosized particles may exhibit unexpected functions different from their original bulk constitution [5]. Recognizing the structural, behavioral, and functional changes identified when a substance’s size is reduced to the nanometric range, and the challenges of NCs stabilization, is an essential consideration for biological utilization. Some NCs are composed of three layers: (1) the surface layer, functionalized with a variety of small molecules, surfactants, metal ions, and polymers; (2) the shell layer, which is constituted by chemically different material from the core counterpart; (3) the core, the central part of the NC often referred to as the NC itself [6]. Due to their exceptional characteristics, NCs can be employed for drug delivery systems, chemical and biological sensing, CO_2_ capturing, gas sensing, and other related applications [7,8,9,10,11].

Nanosized carriers, whether of simple or complex composition, display unique physicochemical properties and represent an excellent tool in the development of nanodevices to be used in biomedical, physical, biological, and pharmaceutical interventions [12]. Of note, NCs have aroused great interest in the medical community for their ability to deliver drugs in a time-dependent and optimal dosage—potentially increasing the therapeutic efficacy of treatment, while reducing side effects and improving patient’s management and quality of life [13]. To produce a less-toxic and more compatible nanocomposite, knowledge of their new molecular behavior in the human organism is essential. Lastly, defining biological and biophysical characteristics such as size, shape, flexibility, surface modifications, and molecular interactions are important to successful clinical administration [14].

Melatonin (N-acetyl-5-methoxytriptamine) is an indole heterocyclic compound secreted by the pineal gland during darkness but also likely released by many cells in a non-circadian manner [15]; melatonin released from non-pineal cells, however, does not enter the blood but functions in an autocrine or paracrine manner. It is an ancient molecule present in perhaps all animal and plants, and has been studied for many biological applications [16,17,18]. Melatonin is a type II agent possessing low water solubility and high permeability, with short half-life and restricted bioavailability when taken orally [19]. In addition to acting as free radical scavenger, antioxidant, and antitumor agent [20,21,22], melatonin efficiently stabilizes cellular membranes, modulating activities of enzymes while improving cellular function. Overall, the functions of melatonin can be receptor-mediated (e.g., through melatonin receptor (MT) termed MT1, MT2, and MT3 receptors) or receptor-independent by directly interacting with intra or extracellular molecules [23].

Melatonin is often administered orally but it presents a variable bioavailability. A systematic study investigated the pharmacokinetics of alternative administration routes of melatonin [24]. Authors documented that intranasal instillation of melatonin has a rapid absorption rate and high availability. Transdermal application of melatonin displayed variable absorption due to its slow release with possible deposition of melatonin in the skin. Oral transmucosal administration of melatonin resulted in high plasma concentration compared to oral administration, possibly due to avoiding first-pass effect of metabolism and allowing a direct absorption into systemic circulation. Subcutaneous injection of melatonin also showed a rapid absorption compared to oral intake but did not exhibit advantages compared to other administration routes. As one example, prolonged release preparations of melatonin have been adopted in Europe and other countries for the acute treatment of primary insomnia in patients older than 50 years [25], displaying short- and long-term efficacy and safety. The sustained-release melatonin based on multiparticulate matrix has been designed to restore the sleep-wake cycle using polymeric pellets by extrusion-spheronization technique [26]; in vivo pharmacokinetics of melatonin delivery showed full release after 8 h of gastrointestinal transit along with low viscosity when associated with hydroxypropymethylcellulose as cushioning agent.

In 2017, a comprehensive study revisited the nanosized platforms including liposomes, ethosomes, niosomes, polymeric nanoparticles (NPs), solid lipid NPs (SLN), and cyclodextrins, to address for melatonin delivery to skin via both in vivo and in vitro applications [27]. Authors reported a higher efficacy of melatonin-loaded NCs than conventionally administered melatonin, mainly due to its protection from premature oxidation and enhancement of drug penetration due to the more appropriate exposure times. For these same reasons, the use of nanoformulated melatonin was also recently proposed as a potential treatment of COVID-19, where the oxidative stress plays a key role [28]. Considering the innovative delivery system for melatonin and the emerging search for the development of new pharmaceutical nanotechnologies, the incorporation of the indole into nanosystems becomes particularly very attractive relative to its therapeutic applications in several diseases. Figure 1 summarizes the potential nanostructures into which melatonin can be incorporated to amplify its effectiveness and safety for therapeutic interventions.

### 1.2. Melatonin: Chemical Modulation and Solubility

There is emerging evidence on the effective use of modified melatonin products in several diseases [29,30,31]. Of note, modification of the acetamide moiety, substitution of the second position of the melatonin indole ring, and substitution of the methoxyl group of melatonin are possible chemical changes that mitigate the effectiveness of melatonin. Recently, a novel melatonin derivative was synthesized by introducing a sulfonate group 4-(3-(2-acetamidoethyl)-5-methoxy-1*H*-indol-1-yl) butane-1-sulfonate (MLTBS). This compound was synthesized from melatonin by treatment with NaH and 1,4-butane sulfone in tetrahydrofuran (THF), and conserved the original function of the indole [32]. In addition to maintaining cell functions similar to those of melatonin, MLTBS has good safety and water solubility both in vitro and in vivo.

Melatonin has a poor stability and solubility in aqueous solution [33]. Although melatonin at dose of 100–113 µg dissolved in 5% ethanol and 95% isotonic saline has shown stability for six months, melatonin (50 µg/mL) dissolved in phosphate buffer was degraded over 21 days following various pH ranges [34,35]. Other chemical solvents for melatonin include dimethyl sulfoxide (DMSO), glycofurol, and propylene glycol. Using HPLC method, glycofurol and DMSO showed sufficient stability for melatonin solutions over 45 days at 25 °C; tested concentrations were of 10 mg of melatonin in 1 mL (20% *w/w* glycofurol and 40% *w/w* DMSO, and 1 mg/g (50% DMSO) [36]. These vehicles may be harmful to the organism due to their induced molecular changes and cytotoxicity. Therefore, the use of novel biocompatible nanomaterials may facilitate the solubilization of melatonin by improving their chemical interaction.

To effectively characterize the real actions of melatonin-loaded NCs, we examined the literature to identify the in vitro and in vivo involvement of these formulations with potential therapeutic applications, highlighting advantages and disadvantages observed in different clinical manifestations of the diseases or conditions.

## 2. Melatonin Incorporated into Lipid-Based NCs/Nanosystems

Melatonin is poorly soluble in an aqueous solution. In this regard, its half-life, bioavailability, and distribution into cells are potentially limited in the biological environment [37]. To improve their solubility and permeability, the loading of poorly water-soluble chemicals into delivery systems based on lipid raw materials was developed to satisfy these requirements; these systems include liposomes, solid lipid NPs (SLN), and hybrid nanocapsules (mixture of lipids and polymers), among others [38] (Figure 2). These lipid-based NCs have unique properties of biocompatibility and a distinct route of absorption, thereby eliminating various physiological barriers for melatonin’s action while providing therapeutic advantages related to the facilities of scalability and industrial adaptability [39].

### 2.1. Liposomes NPs as Carriers of Melatonin

Liposomes are artificial nanoscopic vesicles comprised of an aqueous core surrounded by a concentric phospholipid bilayer, similar to a plasma membrane (Figure 2A). A wide range of potential applications of liposomes is mainly attributed to its ability to incorporate hydrophilic substances in the aqueous compartment, hydrophobic within the lipidic membrane, and amphiphilic molecules in the lipid aqueous interface [40]. Liposomes have been the most popular NCs since they were discovered, and various liposomal formulations are clinically approved [41,42]. A number of techniques have been reported for liposomal preparation, including ethanol injection, reversed-phase evaporation, detergent dialysis, and high-pressure homogenization [43]. However, liposomes have a relatively low loading capacity and poor stability, in addition to being extremely expensive [38,42]. There are recent studies reporting that Zwitterionic polymers are capable of stabilizing the liposomes [44,45,46], since they are formed by equal amounts of anionic and cationic groups on the molecular surface, with high hydrophilicity and antifouling properties.

To overcome their low stability, Gonçalves et al. [47] used the natural polysaccharide chitosan to promote stabilization of melatonin-loaded phosphatidylcholine liposomes; these were named chitosomes. Chitosan improves the stability after 90 days of storage at room temperature compared to liposomes free from chitosan which showed decreased diameter values and polydispersity indexes. In addition, chitosomes presented a reduction in the bilayer thickness, revealing a better-organized NC [47]. Interestingly, ethanolic liposomes (ethosomes) loaded with melatonin and stored for 120 days showed only 7.6% increase in size, suggesting a stabilizing effect of ethanol in the formulation by providing a net negative charge on the surface and avoiding aggregation [48]. Curiously, an in vitro dialysis set-up of melatonin into liposomes had a desirable and slower release profile compared to free-melatonin dissolved in 10% ethanol [41]. This is probably due to the fact that melatonin diffusion from lipid bilayers is hindered by the aqueous medium, acting as a barrier to the melatonin transport.

Zhang et al. [43] identified a new method to improve the effectiveness of melatonin-loaded liposomes. Under supercritical pressure of CO_2_ (140 bar), they obtained better results of drug encapsulation efficiency (EE = 82.2%), and an average liposome diameter of 66 nm, achieving a high bioavailability of melatonin in optimal parameters for therapeutic purposes. Melatonin-loaded liposomes or ethosomes exhibited a potential alternative for a variety of pathological conditions. Considering ethosomes (EE = 70.71% vs 49.2% of liposomes), a better skin drug deposition and tolerability was obtained in the rabbit skin, suggesting that ethosomes may offer a suitable approach for transdermal delivery of melatonin [48]. To enhance the transdermal penetration of melatonin, Marepally et al. [49] developed novel lipids for a nanoparticle system to evaluate improvement in the drug penetration. Among the lipids, the formulation Cy5 (1,1-Di-((*Z*)-octadec-9-en-1-yl) pyrrolidine-1-ium iodide) prepared with ethanol was the most efficient by increasing transdermal permeation of melatonin (134 μg/cm^2^ after 5 h) through the interaction between the components of stratum corneum.

Melatonin also exhibited a beneficial response through the inhibition of oxidative damage in the liver and lungs of rats treated with liposomal or nanoencapsulated melatonin (4.46 mg/kg body weight, intravenous), thereby demonstrating their strong free radical scavenging, antioxidant, and antigenotoxic properties, and liposomal targeting efficiency [50].

### 2.2. Solid Lipid NCs/Nanosystems for Melatonin Delivery

Solid lipid NPs (SLN) are NCs consisting of a solid fat core stabilized by a layer of surfactant molecules on its surface (Figure 2B). They use a mixture of one or more compatible and biodegradable lipids with a melting point above 40 °C, maintaining the solid status at room temperature or at body temperature [51]. SLN are characterized by the presence of a high specific surface area due to their spherical shape, small size, and favorable zeta potential, thus enhancing the system stability [52], and providing a sustained drug release with minimal toxicological risks [38]. A new version of SLN, called nanostructured lipid carriers (NLC) was developed (Figure 2C) using a mixture of solid and liquid lipids to improve their drug-loading capacity and stability [51]. Both SLN and NLC are based on drug incorporation in a melted lipid and mixed with the aqueous surfactant solution. Different methods used in the obtaining of SLN and NLC based on either high energy (high-pressure homogenization, ultrasound techniques, and supercritical fluid technologies) or low energy procedures (solvent emulsification evaporation, coacervation, microemulsion, and phase-inversion temperature method) are followed by immediate cooling leading to lipid recrystallization and NC formation [53]. Due to their characteristics, SLN and NLC have been used as a drug delivery system of poorly water-soluble and lipophilic compounds for different administration routes, such as topical, oral, systemic, ocular, and pulmonary [38].

Through its chronobiotic action, melatonin is used for the control of sleep disorders due to its ability to synchronize the circadian system. To enhance its availability, Albertini et al. [54] obtained SLN (3–6 mg of melatonin) with excellent flow ability and usable as a bulk oral powder, unit dose-filled sachets, or sprinkled onto foods before administration. More recently, Musazzi et al. [55] developed an important delivery system for melatonin by incorporating SLN in orodispersible films (ODFs) made of maltodextrin. Using hydrogenated castor oil as a lipid carrier for melatonin (6.0 mg/cm^2^), they obtained a sustained release over at least 5 h in saliva, gastric, and intestinal simulated media on ODFs containing SLN; the NC showed good suitability to be swallowed without water, allowing the adjustments for drug release according to the clinical needs.

For topical applications, melatonin penetration through the dermal or ocular routes is an important issue. SLN was shown to enhance the photostability of melatonin since photolysis experiments demonstrated an extent of 19.6% degradation for unencapsulated melatonin compared to 5.6% for melatonin-loaded (7 mg) tristearin-phosphatidylcholine SLN [56]. Transdermal drug delivery offers many advantages over conventional methods such as avoiding the first-pass metabolism, reducing systemic side effects, and providing a sustained drug release [57]. With this proposal, melatonin (3 mg)-loaded phosphatidylcholine SLN exhibited a sustained slow release up to 24 h with detectable plasma level after transdermal application [58]. Similar results were obtained by Hatem et al. [59] using NLC loaded with 25 mg of melatonin and using polysorbate 80 as surfactant to treat androgenic alopecia. Due to its sustained release, melatonin-loaded NLC improved skin penetrability potential compared to free melatonin, resulting in increased hair density and decreased hair loss in humans [59]. The therapeutic ocular effects of melatonin were enhanced by its encapsulation in cationic SLN made with stearic acid (SA) as a lipid modifier. The authors observed that intraocular pressure was decreased for an extended period in rabbit eye treated with SA-SLN loaded with 451.1 μg/mL of melatonin, and the positive surface charge was effective to ensure mucoadhesion and longer retention time. The cationic SLN with melatonin is useful for the ophthalmic treatment since a favorable ocular tolerability profile was obtained [60].

Other studies employed melatonin-loaded SLN via systemic administration in order to investigate the recognized antioxidant potential of melatonin against damage caused by exogenous agents, such as cyclosporine A (CsA). In this regard, Rezzani and colleagues observed CsA-induced cardiotoxicity (15 mg/kg/day) due to the promotion of lipid peroxidation and apoptosis in the heart of rats which was reversed in animals that received SLN as a melatonin delivery system for 21 days. Melatonin’s antiapoptotic efficacy, at the dose of 1 mg/kg/day, is mainly observed when it is loaded in SLN, suggesting that they are rapidly absorbed through endocytosis to exert a more effective antiapoptotic process [61]. A potent antioxidant effect of melatonin (single dose of 25 mg/kg) loaded into SLN was also observed by Mirhoseini and colleagues which was able to significantly decrease the levels of malondialdehyde in testis by improving both seminiferous epithelium diameter and thickness in rats exposed to testicular traumatic injury [62]. After being incorporated into NLC, melatonin was effective for the treatment of breast cancer MCF-7 cells. In addition to blocking cell proliferation itself, melatonin-loaded NLC potentiated the effects of the chemotherapeutic agent tamoxifen to induce apoptosis with a marked decrease in anti-apoptotic survivin while increasing pro-apoptotic Bid mRNA levels [63].

### 2.3. Hybrid NCs/Nanosystems Composed of Lipids and Polymers as Carriers of Melatonin

Based on lipid and polymeric NCs limitations, lipid-polymer hybrid NPs (LPN), also called lipid-core nanocapsules (LPC) emerged as a new NC for drug delivery (Figure 2D). They share the characteristics of both lipid (SLN and NLC) and biocompatible polymeric particles [38]. While lipid increases the loading capacity and drug permeation, the polymer regulates the drug release [64]. LPN is structured by diffusion of a sorbitan monostearate in capric/caprylic triglyceride, forming a solid organogel surrounded by a polymer wall, and coated with the polysorbate 80 as a stabilizing surfactant [65]; this system allows the loading of hydrophobic drugs into lipid core and hydrophilic drugs into polymer shell. Although the LPN structure is complex, their preparation methods are quite simple. As a recently developed nanoparticle system, LPNs have been largely used in the latest years [38].

Overall, most studies using LPN as carriers of melatonin have focused on the antioxidant and antiapoptotic effect of this compound in different experimental models, such as liver microsomes [65,66], in vitro antioxidant assay [67], pulmonary cells [68], nematodes [69,70], and bovine embryo culture [71,72]. The presence of melatonin in LPN has decreased significantly lipid peroxidation depending on specific dose and lipid substrate. In this regard, melatonin, at doses of 400 μM, was sufficient to provide their protective effects; the encapsulation rate of melatonin in LPNs (55%) was higher than that of nanoemulsion (33%), thus improving its antioxidative effect [65,73]. Despite this potential, the co-encapsulation of lipoic acid with melatonin (2.4 or 4.8 mM) in LPN did not improve the protection against lipid peroxidation in the liver microsomes compared to the nanoencapsulation of individual agents [67]. As a pre-treatment, melatonin-loaded LPN (0.93 mg/mL, EE of 32.11%) also showed less cytotoxicity and genotoxicity (reduced DNA damage) than free melatonin to protect the alveolar epithelial A549 cell line from the paraquat (PQ)-induced oxidative stress [68]. Likewise, more recently, an in vivo model of PQ-induced toxicity was also used to examine the pharmacological and toxicological effects of melatonin-loaded LPN [69]. In this study, melatonin-loaded LPN enhanced the fluorescence intensity of the transgenic strain that encodes the antioxidant enzyme SOD-3, showing a possible protective mechanism against PQ-induced damage. Furthermore, LPN showed a lethal dose of 50% in the *C. elegans* near to the highest concentration tested (118.50 × 10^12^ LPN/mL), indicating low toxicity of the LPN. Moreover, pre-treatment with melatonin-loaded LPN (0.967 mg/mL) significantly increased the survival rate and development of *C. elegans* exposed to PQ compared to untreated or pre-treated with free melatonin [70].

Melatonin has also been used as a supplement in culture medium to improve the efficiency of in vitro mammalian embryos and oocyte quality [71,72]. Melatonin-loaded LPN with 10^−9^ M drug concentration had the highest hatching rate in bovine embryos, thus increasing the cleavage rates while decreasing cell apoptosis per blastocyst and ROS formation. The embryos showed a downregulation of the pro-apoptotic CASP3, SHC1, and BAX genes while upregulating the anti-apoptotic MCL1, and the expression of GPX1, SOD1, SOD2, and CAT antioxidant genes. Due to the size of the melatonin-loaded LPN (168 nm diameter), it crosses the pores of the zona pellucida and plasma membrane during oocyte maturation, remaining at intracellular sites until the blastocyst stage [71].

To enhance the stability and interaction of LPN with biological membranes, surface modifications with cationic polymers or lipids have been proposed. Carbone et al. [74] examined hybrid versus polymeric nanocarriers produced by a low-energy organic-solvent-free method after loading with melatonin (0.03% *w/w*) for ocular administration. They reported favorable coating properties clearly influencing the physical properties of the systems in terms of mean particle size, surface charge, shape, and stability. Due to its non-spherical shape and sustained/prolonged drug release (EE 87.69%), dimethyldioctadecylammonium bromide surfacing was the most suitable coating material for the NCs, thereby representing a great advantage for ophthalmic application.

## 3. Non-Ionic Surfactant-Based Vesicles (Niosomes) for Melatonin Delivery

Niosomes are nanostructures constituted by non-ionic surfactant-associated membranes, highly stable but slightly leakier compared to the liposomes; their permeability for KCl ions is elevated and the size of niosomes decreases upon freezing [75]. Niosomes are unilamellar or multilamellar self-assembled vesicular NCs that serve as a suitable carrier of hydrophilic and lipophilic drugs with broad applications from dermal delivery to brain-targeted delivery [76]. Since niosomes are manufactured with simple methods, low cost, and stability over extended period, they offer an excellent alternative for drug-delivery system in lieu of the liposomes.

Melatonin-associated niosome gel improves the pharmacokinetics of exogenous melatonin [77] (Figure 2E). The ex vivo residence time of melatonin-based niosomes showed more than 3 h maximum adhesiveness at 37 °C. Randomized double-blinded healthy volunteers (n=14) tested the efficacy of topical oral transmucosal melatonin-niosomes gel at doses of 2.5, 5, and 10 mg; the higher absorption and prolonged systemic circulation of melatonin revealed an efficient method of systemic delivery with dose-proportional pharmacokinetics. More recently, the same research group showed the potential use of glutaryl melatonin (14.2 mM) nanoformulated as niosome gel against 5-fluorouracil-induced oral mucositis in mice [78]. They concluded that glutaryl melatonin-loaded niosomes produced mild anti-candidiasis and anti-inflammatory activities, with higher fungicidal effect than that of melatonin-loaded niosomes. Uthaiwat et al. [79] confirmed that melatonin niosome gel also inhibits inflammation and lipid oxidation in a model of 5-FU-induced oral mucositis.

Melatonin encapsulated in niosomes was preclinically evaluated through pharmacokinetic, pharmacodynamics, and toxicity assays in male Wistar rats. Based on therapeutic doses, intranasal melatonin niosomes had no adverse effects to animals and provided sustained delivery to induce sleep with delayed systemic circulation in peripheral tissues compared to intravenous administration of melatonin [80].

Niosome-incorporated melatonin has also been studied for preventing or treating ultraviolet (UV)-induced skin damage; since while conventional chemical UV filters (sunscreens) accumulate on upper skin, melatonin can penetrate deeper. By preparing these melatonin-loaded niosomes with Tween80/Span80 mixture (146 nm, PI = 0.438), an entrapment efficiency of 58.42% was observed. Ex vivo analysis in the rat skin showed 58% of melatonin permeation vs 7.4% of octylmethoxycinnamate (a traditional sunscreen formulation), and 37% of melatonin accumulated in the skin after 24 h exposure. Thus, this combined nanoformulation exhibited high antioxidant activity like melatonin itself, thus revealing an optimized emulsion with co-delivery purposes [81].

## 4. Melatonin-Loaded Silica-Based NPs

Inorganic nanomaterials (e.g., silica) have important properties associated with nano-scale dimensions, and serve as a drug delivery system for the treatment of multiple pathologies including cancer [82]. Due to their high surface-to-volume ratio, molecular agglomeration is a common feature. To overcome this problem, the surface of inorganic NPs can be functionalized with polymer chains by either chemical or physical interactions [83]. Since silica NPs have unique optical properties, high surface area, adsorption and encapsulation abilities, high biocompatibility, and low toxicity, they are used in multiple applications such as diagnosis and therapy of diverse biomedical conditions, genetic corrections, biosensors development, enzyme supporters, and cellular uptake [84].

The in vitro behavior of thymoquinone-melatonin (TQ-MLT) was examined in targeted drug delivery system consisting of silica NPs and modified with diamine polymer, carboxymethyl-β-cyclodextrin, and folic acid. The modification with diamine polymer showed peaks at 2900–2950 cm^−1^ vs non-modified NPs, and a more intense peak of 3300 cm^−1^ was identified in longer polymers through the characterization by Infrared Spectroscopy, suggesting the formation of more hydrogen bonds. In vitro assay was performed with five suspensions of silica NPs (0.05 mg/mL) incubated with HeLa cells for 24 h. The release of TQ-MLT increased as the polymer length decreased; shorter polymers had massive burst release, with the discharge of large amounts of drugs released within 1 h exposure. Longer polymers presented a more efficient sustained and pulsatile release of melatonin within the first 5 h, and HeLa cell toxicity was increased with the polymer length [85] (Table 1).

## 5. Graphene and Melatonin Delivery

Graphene is a carbon-derived nanomaterial used in several applications, especially (but not only) in biosensing and electrochemistry, due to its unique structure with electronical, thermal, and mechanical properties [100]. The derivatives include graphene oxide (GO) and reduced graphene oxide (rGO) equally classified according to the number of layers, but differing in their surface chemistry/dimension [101]. Because GO is hydrophilic, soluble, and stable in colloids, it has the ability to serve as drug delivery system, being potentially functionalized with other molecules [102].

To evaluate its therapeutic efficiency in cancer cells, a functionalized graphene-dendrimer system was produced with Fe_3_O_4_ NPs as magnetic NC for co-delivery of melatonin and doxorubicin to human osteosarcoma cells [Saos-2 and MG-63, and Human Bone Marrow Mesenchymal Stem Cells (hBM-MSC) lines]. The platform resulted in many functional chemical groups (-OH, -COOH, and -NH2) after modification of β-cyclodextrin grafted with graphene oxide. Encapsulation efficiency values were 21.5% for melatonin, and a good antitumor performance was observed of this graphene-dendrimer system as magnetic NC, which was confirmed by cellular uptake and apoptosis induction (Table 1). By combining melatonin with doxorubicin, an important synergism occurred mainly related to the downregulation of X-linked Inhibitor of Apoptosis, survivin, and human telomerase catalytic subunit. This graphene-associated nanosystem had anticancer efficacy against osteosarcoma while exhibiting low toxicity in normal cells [86].

## 6. Nanofibers and Nanocapsules as Biomaterial for Melatonin Controlled Release

Since antiquity, wound healing has been a challenge for maintaining physical integrity and good health. The process of wound healing is rather complex, involving different cell types, cytokines, minerals, growth factors, and vitamins [87]. Advances in medical sciences has created a large amount of knowledge for the development of a specific branch devoted to wound healing [103]. Tissue engineering has been developed in response to the shortcomings associated with cells that are lost due to disease or trauma. The driving force behind tissue engineering is to produce biological constituents capable of replacing the damaged tissue [104]. For this purpose, Mirmajidi and colleagues developed and tested chitosan (Cs)-polycaprolactone (PCL)/ polyvinylalcohol (PVA)-melatonin/chitosan-polycaprolactone three-layer nanofiber wound dressing, based on electrospinning for sustained release of melatonin. The wound healing effect was analyzed using a full-thickness excision model of rat skin under local administration of melatonin. It was observed a complete regeneration of the epithelium, associated with an efficient collagen synthesis, wound remodeling, and reduction of inflammatory cells (Table 1). For clinical application, this wound dressing might be used as a promising drug nanosystem in different wound types, such as in diabetic ulcers, trauma, and burns [87].

Li et al. [88] explored the potential use of bacterial cellulose (BC) for oral delivery of 2 mg/kg melatonin to improve its oral bioavailability and low solubility. BC was hydrolyzed with sulfuric acid followed by oxidation to prepare bacterial cellulose nanofiber suspension (BCNs). Melatonin-loaded BC nanofiber suspension (MLT-BCNs) was prepared by emulsion solvent evaporation method. Interestingly, the nanofibers of BC became short and thin compared with non-nanoformulated BC. Melatonin was equally distributed in the MLT-BCNs, both exhibiting great thermodynamic stability. The MLT-BCNs showed a more rapid dissolution of melatonin compared to free melatonin; the cumulative release rate was about 2.1 times higher than that of commercial melatonin. The bioavailability of MLT-BCNs was about 2.4 times higher than the MLT in free form. Thus, MLT-BCNs acted as a promising delivery system in the rat model with enhanced dissolution and bioavailability of melatonin after its oral administration.

Bessone et al. [89] documented the synthesis and application of a nanosystem for the controlled release of melatonin in the retina of rabbits. The ethylcellulose nanocapsules were studied by zeta potential, scanning electron microscopy, hydrodynamics, and for in vitro releasing analysis. In vivo experiments, using an albino rabbit’s model of retinal degeneration, showed a complete trans-corneal permeation. An important observation was the slower release of melatonin (1 and 2 mg/mL) when transported by nanocapsules. Melatonin-loaded ethylcellulose nanocapsules showed high corneal penetration in ex vivo and in vivo analyses, and were primarily related to its neuroprotective effect on retinal ganglion cells (Table 1). These findings emphasize the need for novel nanotechnologies in the treatment of neurodegenerative diseases at the ocular level.

To improve the release of melatonin, Vlachou and colleagues generated the electro spun-melatonin loaded nanofibers, formulated with hard gelatin and DRcapsTM capsules, whose fiber matrices were prepared with cellulose acetate (CA), polyvinylpyrrolidinone (PV), and hydroxypropylmethylcellulose (HP). The modified release of melatonin was evaluated based on these three matrices using gastrointestinal fluids at different pH conditions. The nanofibers, namely CA1, CA2, PV1, HP1, HP2, and nanocomposite formulations of CAPV1-CAPV5 in hard gelatin capsules, showed fast melatonin release at pH of 1.2. When encapsulated in DRcapsTM capsules, the nanocomposite CAPV1 and CAPV2 delivered 52.08 % and 75.25 % of melatonin, respectively, at a slower pace for 6 h. The other nanofiber formulations were capable of delivering 100% of melatonin within 6 h. These melatonin-loaded nanofibrous mats displayed a promising deliver profile for the treatment of sleep dysfunctions [90]. The same authors created monolayer and three-layer tablets, incorporating nanofibrous mats composed of CA and PV loaded melatonin to examine time release. In vitro dissolution of melatonin showed pressure- and pH-dependence in gastrointestinal-like fluid. The release of melatonin from physical mixture of tablets was relatively slower than that of nanofibers-based tablets, revealing important properties for the control of sleep-onset and the maintenance dysfunctions (Table 1). Since the three-layer tablets are composed of hydroxypropylmethylcellulose or lactose monohydrate, the slow release of melatonin can be adjusted according to its chronobiotic profile, showing a closer alignment to the natural availability compared with other delivery systems [91].

An increased efficacy of peripheral nerve regeneration was achieved by using a biodegradable porous neural guidance conduit as a NC for transplantation of allogeneic Schwann cells (SCs). Salehi et al. [92] prepared their conduits with polyurethane (PU) and gelatin nanofibers (GNFs) using thermal separation technique, and filled with melatonin and platelet-rich plasma. The fabrication enhanced the proliferation of SCs. After the conduit was seeded with 1.50 × 10^4^ SCs (PU/GNFs) plus melatonin and platelet-rich plasma, it was implanted into a 10-mm defected sciatic nerve of Wistar rat for functional assays. The combination of nanofibers with melatonin and plasma ameliorated nerve functional index, muscle action potential amplitude and latency, hot plate latency, weight-loss of wet gastrocnemius muscle, and histopathological features, highlighting the enhanced regenerative capacity of the nanoformulation (Table 1).

## 7. Chitosan-Based NPs for Melatonin Delivery

Chitosan is considered a semisynthetic polyaminosaccharide generated by N-deacetylation of chitin. Chitosan NPs are well described NCs for drug, gene, and protein delivery [105]. The most common chitosan formulation includes ionotropic gelation through interaction of positively-charged aminosugar monomers and negatively-charged polyanions (e.g., tripolyphosphate, hexametaphosphate or dextran sulfate) [106,107]. There are some biochemical barriers to be overcome to exploit the clinical potential of chitosan-based NPs, such as their instability, bioavailability, and toxicity [108]. Regarding their toxicity, the lecithin/chitosan NPs did not cause any changes in the plasma membrane and cell viability neither at concentrations of 400 μg/mL in Caco-2 epithelial cells [98] or in brain tumor U87MG cell line [93], nor at 200 μg/mL in HaCaT human keratinocyte cell line and BJ fibroblast cells [97]. However, concentration-dependent chitosan cytotoxicity has been demonstrated in a free soluble form, thus indicating that this form of chitosan is much more cytotoxic than when it is incorporated in the nanosystems [94,109].

The main material that is usually combined with chitosan in melatonin-loaded NPs formulation is lecithin, which is a natural lipid mixture of phosphatidylethanolamine and phosphatidylcholine used as a compatible excipient. The association of negatively charged lipid material (e.g., lecithin) with a positively charged polysaccharide (e.g., chitosan) enhances the bioadhesive and penetration properties of NPs. The wound healing potential of lecithin/chitosan NPs loaded with melatonin was shown in in vitro [94,110] and in vivo studies [95]. The positively charged lecithin/chitosan NPs also improved the melatonin release and permeability, overcoming the limitations of the short half-life of free melatonin by extending its residence period [94,95,98,109] while increasing the therapeutic efficacy of melatonin in growth inhibition of human glioblastoma U87MG cells [93]. The association of chitosan and tripolyphosphate was used in melatonin-loaded NPs against the etoposide-induced genotoxicity effects in HepG2 cells [96]. Nanoformulated melatonin was more effective than free melatonin for reducing the deleterious effect of etoposide via lowering DNA damage and oxidation (Table 1).

The therapeutic potential of chitosan-based NPs loaded with melatonin is primarily dependent on long-term stability during storage and encapsulation (entrapment) efficiency (EE), which affect the amount of drug that is released during the treatment. Since melatonin is susceptible to degradation in contact with air and light, and has low solubility in water, its encapsulation in NPs may prevent degradation while increasing its photostability [56]. Hafner et al. [111] examined the long-term stability of lecithin/chitosan-based NPs after different processes of lyophilization. The optimal results were presented by the lyophilisates with trehalose (2.5% *w/v*), which was capable of retaining the physicochemical properties of melatonin-loaded NPs following 7 months of storage at 4 °C. Similar results were obtained with lecithin/chitosan-based melatonin-loaded NPs without lyophilization after 28 days of storage at 4 °C [95]. Furthermore, chitosan lipid enriched NPs improved conservation of these nanosystems for six months at 5 °C [110]. A new method to maintain the organic properties of fruits employed chitosan-incorporated melatonin; in this method, melatonin improved both antioxidant and antimicrobial activities of the nanocomposite when it was assembled layer-by-layer [112].

In general, a great variation is observed in the EE [(actual drug content/theoretical drug content) × 100 = EE (%)] among the chitosan-based NPs formulations. El-Gibaly [99] observed that EE of melatonin into microcapsules decreased as the concentrations of sodium lauryl sulfate (NaLS) and dioctyl sulfosuccinate (DOS) increased to obtain buoyant microcapsules as a vehicle for drugs administered orally. However, chitosan buoyant microcapsules prepared with a drug/polymer ratio of 4:1 (EE 56.23%) and NaLS–DOS combination in a 1:2 ratio showed better efficiency when compared with free melatonin solution in terms of drug release profile (50% in 5 h). This author reported that encapsulated melatonin promoted antiapoptotic activity in rats exposed to aflatoxin B1, serving as an alternative therapy in cases of aflatoxicosis (Table 1). Chitosan-based NPs formulated with lecithin S45 (20:1 ratio) presented an EE of 26.0 to 38.2% and slower melatonin release of 50% in 1.7 h with a maximum release of ~70% until 9 h [98]. More recently, Lopes Rocha Correa and co-workers, using a lecithin-chitosan ratio of 1:1, obtained EE of 27% (108 μg/mL), which corresponded to a similar amount of melatonin encapsulated in NPs using 20:1 lecithin-chitosan ratio (95.7–109.2 μg/mL) [98], but with a slow drug release of 42.9% in 24 h [95] (Table 1). Better results were obtained with the use of chitosan/tripolyphosphate NPs, for which the EE of melatonin was 75% and the release after 7 h was 87% [96]. Overall, these studies reported that chitosan-based melatonin-loaded NPs are more effective than melatonin in its free form.

## 8. Synthetic Polymeric NPs as Carriers of Melatonin

Another NC used to improve therapeutic efficiency of different molecules includes the polymeric NPs. These nanomaterials favor targeted drug delivery and increase the time of circulation in the organism, with minimum side-effects; polymeric NPs (sized 10–1000 nm) sometimes can accumulate into cells without inducing phagocytosis [113]. Polymeric NPs are biodegradable NCs, protecting a drug that is either adsorbed or chemically linked to the surface or even encapsulated within the polymeric nanostructure. The possibility to synthesize polymers with well-controlled structures and composition supports the fine-tuned properties of the NPs for pharmacological application [114]. There are multiple types of polymers used for the synthesis of NPs as carriers of melatonin (e.g., hydrophobic, hydrophilic, etc.). Each kind of polymer gives rise to NPs with different features such as size, morphology, surface charge, polydispersity index, encapsulation efficiency, stimuli responsiveness, functionalization capability, among others. These physicochemical properties directly influence the pharmacokinetic and pharmacodynamic behavior of NPs. In addition, not all drugs may be loaded onto any kind of polymer. Therefore, the choice of the most suitable polymer at the time of producing NPs will depend both on the therapeutic purposes to be pursued and on the characteristics of the drug to be loaded [115]. In this particular case, physicochemical properties of melatonin, as well as the pharmacotherapeutic objectives of its use, will significantly condition the selection of the most appropriate polymers for manufacturing NPs which are able to load this indoleamine. The synthetic polymers discussed herein include polycaprolactone, poly-lactic acid, poly (lactic-co-glycolic acid), polyethylene glycol, and poly (methacrylic acid-co-methyl methacrylate).

Using different polymers, Schaffazick et al. [116] studied the stability of nanocapsules with melatonin (1.5 mg/mL) constructed by interfacial deposition. The melatonin’s nanoemulsion and nanodispersion showed stability (nanoparticulated systems between 134 and 325 nm), and associated melatonin concentration ranged from 29% to 50%. The stability of the NCs was examined for several parameters, and it was observed that depending on the nature of NPs and condition of storage, the melatonin stability may be significantly changed. Figure 3 depicts major melatonin benefits after being incorporated into polymeric NPs and polymeric nanogels.

### 8.1. Polycaprolactone/Melatonin

Melatonin-loaded polycaprolactone (PCL) NPs have been recently reported by de Oliveira Junior et al. [117] as an effective nose-to-brain delivery system of melatonin to treat glioblastoma. These NPs was prepared by nanoprecipitation method and showed increased drug water solubility ~35% fold. Intranasal administration of this nanoformulation resulted in rapid translocation of the NPs to the rat brain (higher AUC brain and drug targeting index). Moreover, melatonin-loaded PCL NPs enhanced cytotoxicity against U87MG cells (IC_50_ = 2500 fold lower vs free drug). Due to its selective antitumor activity, this nanoencapsulation is suitable for the treatment of glioblastoma [117].

Another very useful method for obtaining PCL NPs is the electrohydrodynamic technique of electrospraying [118,119,120,121]. In this regard, melatonin has also been encapsulated in PCL NPs by this method to improve the therapeutic efficacy of synthetic tissue grafts. For melatonin encapsulation, a 3 wt % PCL solution was used, and particles with diameter size of 2.3 ± 0.64 μm were obtained (EE: 73%). These NPs allowed a prolonged release of melatonin for about 8 h. By analyzing in vitro (primary human osteoblasts cells) and in vivo (female Sprague Dawley animals) behavior of grafted materials, Gurler and colleagues identified a remarkable increase in both cell population and bone volume of rats, indicating melatonin-loaded PCL NPs as an optimal electrospray method to improve the therapeutic efficacy of synthetic engraftment [122].

PCL NPs have also been used for developing a transdermal delivery nanosystem of melatonin. The effective encapsulation in the solid phase was confirmed with an EE higher than 80%. The transdermal release of melatonin from the functionalized compound was performed using a synthetic membrane, and the NPs were uniformly distributed on cotton fibers. A continuous and controlled release of melatonin was observed, the kinetics of which were described by the Baker-Lonsdale model [123].

Other melatonin-loaded NCs were prepared by interfacial deposition using amphiphilic diblock copolymer, poly (methyl methacrylate)-block-poly (2-(dimethylamine) ethyl methacrylate), PMMA-b-PDMAEMA, combined with PCL, to monitoring drug release over targeting cells [124]. The transmission electron microscopy analysis showed an oily core recovered by a thin layer composed by PCL/PMMA-b-PDMAEMA. Furthermore, a spherical nano-object with a diffuse polymer corona was observed with an encapsulation efficiency of 25%.

To reconstruct the original structure of the tendon-to-bone insertion site (enthesis) in rotator cuff repair, Song and colleagues produced melatonin-loaded aligned PCL electro spun fibrous membranes. They detected a sustained release of melatonin capable of stimulating chondrogenic differentiation of human bone marrow-derived mesenchymal stem cells (hBMSCs) into chondroid pellet. After melatonin-loaded PCL membranes were implanted in a rat acute rotator cuff tear model, significant recovery properties were reported including inhibition of macrophage infiltration in enthesis interface, augment in chondroid zone formation, promotion of collagen maturation, and reduction of fibrovascular tissue deposition. Since this nanocomposite improved the biomechanical strength of the regenerated enthesis, it may be considered for potential clinical application in the treatment of tendon-to-bone diseases [125].

### 8.2. Poly-lactic Acid/Melatonin

Using HPLC equipped with photodiode array detector, melatonin was validated and determined in poly-lactic acid (PLA) NPs [126]. The in vitro release of melatonin was analyzed in mobile phase consisting of acetonitrile: water (65:35, *v/v*) at a flow rate of 0.9 mL/min, using an isocratic mode and PDA detector at 220 nm. After being studied for precision, linearity, selectivity, limits of detection and quantification, melatonin concentration had an analytical curve ranging from 10 to 100 μg/mL, limit of detection of 25.9 ng/mL, and limit of quantification of 78.7 ng/mL. The intra- and inter-assay coefficient of variation was about 2%, and mean recovery for melatonin was 100.47%. This method was considered safe and suitable for determining melatonin encapsulation efficiency in PLA NPs and can be used to verify the in vitro release of melatonin from different NCs.

The effect of pure melatonin was compared with melatonin entrapped PLA NPs on the reactive oxygen species (ROS), and blastogenic responses in the proliferation of splenocytes obtained from adult golden hamster [127]. The melatonin-loaded PLA NPs was generated by emulsification/nanoprecipitation method, and their physicochemical properties showed good results regarding shape, size, EE (diameter of 36 ± 8 nm with ~78% entrapment efficiency), smooth surfaces, and homogeneous distribution of particle size. The inclusion of melatonin, at the dose of 5000 pg mL^−1^, reduced significantly the size of the NPs. The properties of melatonin-loaded PLA NPs were superior to pure melatonin to improve the immune response by increasing blastogenic response of the splenocytes while reducing ROS formation.

### 8.3. Poly (lactic-co-glycolic Acid)/Melatonin

Poly (lactic-co-glycolic acid), also known as PLGA, represents a biodegradable polymer possessing advantageous properties for the development of NPs and other NCs as drug delivery systems [128]. These advantages include biocompatibility, approval in drug delivery for parenteral route, composition adapted to associate different types of drugs (e.g., hydrophilic or hydrophobic small molecules), protection against drug degradation, prolonged self-life, sustained drug releasing, and ability to be modified with surface-related molecules to provide better interaction with other biological materials while targeting NPs to specific cells [129]. PLGA has been used for the development of drug delivery NCs useful to treat different diseases or improving biological conditions such as cancer, inflammation, vaccination, immune therapy, and other processes that need a minimal dosage of administered drugs [130].

To test the efficacy of PLGA-based NPs loaded with melatonin coated or not with polysorbate 80 (PLGA-PS80), the in vitro antioxidant activities and cytotoxicity were evaluated over erythrocytes [131]. After melatonin was adsorbed in PLGA through an emulsion-solvent evaporation method, physicochemical analyses were performed, and the cytotoxicity of the erythrocytes was assessed by a hemolysis assay. Compared to free melatonin, melatonin-loaded PGLA showed no hemolysis-related action, and scavenging activity of NPs facing ABTS·+ (oxidant agent) was superior to that of free melatonin. Therefore, PLGA and PLGA-PS80 NPs would be promising carriers to improve the in vitro antioxidant effect of melatonin, without producing hemolysis.

PLGA and microparticles prepared with the addition of 0.2% (*w/v*) melatonin have been successfully used to treat osteosarcoma. A sustained release of melatonin was observed under a biphasic pattern with an initial burst in 24 h followed by a sustained release of 70% melatonin up to 40 days. In osteosarcoma MG-63 cells, melatonin-loaded PLGA was uptaken, inducing a cytotoxic effect. Curiously, when melatonin (~1.7 μg) was incorporated in PLGA microparticles, a more appropriate inhibitory effect on osteosarcoma cells was observed, compared to PLGA NPs. The information obtained from this study provides an expectation for incorporating melatonin into polymeric micro/nanocarrier systems as an adjunct for osteosarcoma chemotherapy [132]. Moreover, to verify the osteogenic activity of melatonin and bone morphogenetic protein 2 (BMP-2)-loaded PLGA, Jarrar and colleagues tested the in vitro treatment of pre-osteoblastic MC3T3-E1 cells with both agents either individually or in combination. They found that a higher number of cells and bone mineralization were detected following dual release of low doses of BMP-2 (~20 ng/scaffold) and melatonin (~10 µg/scaffold) [133].

To demonstrate the role of melatonin in an experimental model of sepsis, PLGA [NP-A]) and polyethylene glycol-co-(poly-D,L-lactide-co-glycolide) (PLGA-PEG [NP-B]) were used to examine the loading of melatonin (10 mg/kg) for controlled release. Li Volti and colleagues evaluated oxidative stress in tissue homogenates from septic animals through the expression of heme oxygenase-1 (HO-1), total thiol groups, and lipid hydroperoxide (LOOH). The results showed that both melatonin nanoformulations tested restored thiol group levels, and reduced LOOH levels compared to control sepsis-induced animals. However, the NP-B formulation exhibited higher depletion of LOOH levels in tissues such as heart, lung, and liver when compared with NP-A. Additionally, NP-B formulation significantly reduced HO-1 expression compared to free melatonin, with the exception of the kidney tissue. Due to its improved ability of drug delivery, melatonin-loaded NP-B revealed higher antioxidant activity during sepsis than NP-A [134].

A modified emulsion-diffusion-evaporation method was used to prepare melatonin incorporated PLGA NPs for application during cerebral ischemia-reperfusion insult. Free melatonin or melatonin-loaded PLGA were given orally 24 h before female Sprague-Dawley rats were made ischemic by bilateral clamping of the common carotid arteries. In this study, Sarkar et al. [135] observed that unlike free melatonin, melatonin-loaded NPs demonstrated higher antioxidant potential in brain tissue, even at much lower concentrations. Melatonin-loaded PLGA NPs rescued neuronal cells and mitochondrial activities during cerebral ischemia-reperfusion damage, revealing a novel drug delivery system for protecting brain tissue.

### 8.4. Polyethylene Glycol/Melatonin

The use of polymeric NPs has been applied in the field of stem cell transplantation. To encapsulate melatonin, Ma and colleagues constructed PLGA-monomethoxy-poly-(polyethylene glycol [PEG]) (PLGA-mPEG) NPs. They tested whether the protective effect of melatonin-loaded PLGA-mPEG on adipose-derived mesenchymal stem cells (ADSCs) was improved compared to melatonin in its free form. Melatonin-loaded NPs showed an EE of 79.3%, which reduced the in vitro formation of p53-cyclophilin D complex, preventing the opening of the mitochondrial permeability transition pores, and rescuing ADSCs from hypoxia/reoxygenation injury. Notably, the treatment with melatonin-loaded NPs showed higher ADSC survival rates after rat myocardial infarction than the treatment with free melatonin, supporting the promising therapeutic role of melatonin-loaded NPs in stem cell transplantation for myocardial infarction therapy [136].

PEG-NPs carriers of melatonin have been employed to alleviate sepsis-induced liver injury. To evaluate the effective delivery of melatonin at the damaged site, Chen et al. [137] hypothesized that ROS-responsive NPs formed via the self-assembly of diblock copolymers of PEG and poly (propylene sulfide) (PPS) might potentiate the melatonin’s action. Melatonin was encapsulated by PEG-NPs by using oil-in-water emulsion technique and showed a sustained release that was modulated by in vitro concentrations of hydrogen peroxide. Because melatonin-loaded PEG-PPS NPs had great biocompatibility and produced higher antioxidant effects than free melatonin, the anti-inflammatory effects associated with liver injury were more significantly attenuated by nanoencapsulated melatonin compared to its non-encapsulated equivalent. Therefore, this ROS-mediated on-demand drug delivery method may be considered to enhance melatonin bioavailability during the treatment of oxidative stress-associated diseases such as acute liver injury.

### 8.5. Poly (methacrylic acid-co-methyl methacrylate)/Melatonin

Poly (methyl methacrylate-co-methacrylic acid) or copolymer poly (MMA-co-MAA) is a biocompatible polymer which does not produce toxic residues when is solubilized in water. This copolymer is similar to that of Eudragit S100^®^, a biomaterial approved by the FDA and used as excipient to generate drug tablets to treat various diseases [138]. After melatonin (1 or 10 mg/kg) was loaded into polysorbate 80-coated poly (MMA-co-MAA) NPs and intraperitoneally administered, it reduced lipid peroxidation levels in mice brain (e.g., frontal cortex and hippocampus) and liver tissues compared to free melatonin aqueous solution. Melatonin-loaded NPs also ameliorated the total antioxidant activity in the hippocampus, thereby proving its ability to improve the antioxidative actions of melatonin [66]. The aspects of drug-loaded and drug-unloaded nanocapsules were respectively: low polydispersity (0.01, 0.09), submicronic size (241 ± 55 nm, 207 ± 44 nm), negative zeta potential (33 ± 0.3 mV, 35 ± 1.1 mV), and acid pH (4.2, 4.3). The total content of encapsulated melatonin was 0.996 mg/mL.

### 8.6. Polymeric Nanogel (Hydrogel) for Melatonin Delivery

An innovative study by Atoufi et al. [139] revealed a novel-designed injectable thermosensitive PNIPAM/hyaluronic acid hydrogels containing great amounts of chitosan-g-acrylic acid coated PLGA (ACH-PLGA) NPs to facilitate cartilage regeneration. The ACH-PLGA was used in the hydrogels to improve their mechanical properties, and thus to mimic “natural” cartilage, controlling the release of melatonin as a chondrogenic molecule. The hydrogel showed an optimal integration with the natural cartilage, while presenting an interconnected porous structure. After the mesenchymal stem cells were encapsulated inside the hydrogel, an improvement in cell growth and proliferation occurred. Further analysis showed that melatonin associated with NPs promoted an increase in glycosaminoglycan synthesis; reinforcing the role of the thermosensitive injectable hydrogel as a promising platform for cartilage tissue engineering due to its suitable drug release rate, low syneresis, increased mechanical strength, and bio-adhesion properties.

The development of a melatonin-based nanogel was essential for wound healing improvements. To carry this study, Soriano et al. [140] prepared the nanogel using the following components: poloxamer 407, chitosan, and hyaluronic acid. The nanogel was characterized through physical interactions, wettability, swelling, and internal structure. The release of melatonin followed first order kinetics, and wound healing efficacy was evaluated in animal models with optimal responses. Since melatonin nanogel promoted epidermal growth with evident wound contraction, it can be considered as an efficient drug delivery system to facilitate wound healing process.

## 9. Metallic NPs and Melatonin Delivery

Metallic NPs are naturally synthesized from Ce, Ag, Au, Pt, Pd, Cu, Ni, Se, Fe, or their oxides, mainly via ionic reduction by biological materials or organisms (e.g., yeast, fungus, bacteria, and plant extracts). These nanomaterials have a variety of applications such as in the release of chemotherapeutic drugs, NP excitation by radiation, or even by apoptotic induction [141]. Interestingly, metallic NPs diffuse through the blood vessel endothelium to target specific cells when attached with a ligand to recognize cell receptor, or excited by a magnetic field, X-rays, ultrasounds, and laser light [141]. Of interest, indole molecules (e.g., melatonin, dopamine, epinephrine, and norepinephrine) present low limit of detection when loaded into gold NPs with excitation wavelength of 785 nm, mainly due to the affinity of these NPs for the indole ring [142].

The cellular uptake through the phagocytosis of gold NPs has been demonstrated in murine macrophage cells [143]. Different biocompatible molecules (e.g., melatonin) were added to NPs as stabilizers for gold colloids. Under stirring, aqueous solution of melatonin (0.34 mM) was added for 24 h, and particles were purified by centrifugation. Under electronic microscopy, few melatonin-coated gold NPs were observed in phagosome vesicles near to the nucleus membrane and enclosed in cell podosomes (Table 2).

Gold, palladium, and selenium NPs-associated melatonin has been reported to exert superior efficacy than melatonin alone in experimental studies (Table 2). The gold (Au^+3^)/melatonin complex (35 to 100 nm) was synthesized and inoculated in rats for 30 days to demonstrate protection against testicular damage. While melatonin showed oxidative stress protection, Au^+3^/melatonin combination profoundly decreased the TNF-α and IL-1β levels, and lipid peroxidation, and enhanced antioxidant capacities, thus improving testicular histological aspects and reproductive function [144]. In vitro chemotherapeutic potential of melatonin-loaded Palladium (Pd) NPs was examined in combination with anticancer drugs in lung epithelial adenocarcinoma A549 cells. The Pd-NPs (concentrated at 2.5 μM) incorporated with melatonin at doses of 0.75 mM (ranging from 0.25 to 2.5 mM) resulted in cell toxicity with high leakage of lactate dehydrogenase, intracellular protease, and reduced membrane integrity. Considering the mitochondria, Pd-NPs-melatonin combination decreased the mitochondrial membrane potential, ATP content, mitochondrial number, and mitochondrial biogenesis. By inducing apoptosis and oxidative DNA damage to lung cancer cells, Pd-NPs-melatonin are potentially effective as therapeutic in a higher proportion compared to free melatonin; low concentrations of Pd-NPs and melatonin are well tolerated, and exhibit less adverse effects [145]. Voltammetry determination of melatonin can be achieved through a sensor based on modified carbon paste electrode with Al_2_O_3_-supported Pd NPs. In this work, Soltani et al. [146] described that the limit of detection for melatonin was 21.6 nmol L^−1^ and may serve to detect the indole in human serum and drug samples.

There is sufficient evidence that overproduction of reactive oxygen and nitrogen species leads to acute or chronic liver injury induced by virus, alcohol, lipopolysaccharide (LPS), and other chemicals [147]. Using a Bacillus Calmette–Guerin (BCG)/LPS-induced liver injury mice model, Wang et al. [148] studied the involvement of selenium (Se) NPs-loaded melatonin orally administered at doses of 5, 10, and 20 mg/kg prior to LPS induction. The load of melatonin in Se-NPs reduced the activity of aminotransferase, the extent of hepatic cell damage, and migration rate of inflammatory cells. In addition to attenuating lipid peroxidation and enhancing antioxidant activities (SOD and GPx), Se-NPs-melatonin (10 mg/kg) presented higher hepatocellular protection than free melatonin administered at the same dosage. This nanocomplex showed no side effects to animals, and revealed potent antioxidant properties in the liver.

## 10. Melatonin-Associated Metallic and Non-Metallic Nanocomposites (NCP)

The development of new nanodevices as alternative to target extracellular plaque deposits of the β-amyloid peptide (Aβ) and associated neurofibrillary formations are expected to treat Alzheimer’s disease (AD). A dopamine-melatonin-based NCP possessing synergistic near-infrared (NIR) responses was generated by noncovalent interactions between melatonin (1 mM), dopamine (1 mM) derived quinone, and indole intermediates. After NCP was prepared and characterized by spectroscopy, photothermal conversion, capability, and NIR responsive, it was assessed in both in vitro and ex vivo experiments. Notably, photothermal activity and release of NIR-activated melatonin inhibited Aβ nucleation and self-seeding, in addition to disrupting preformed Aβ fibers during amplification assays of Aβ aggregation. Also, dopamine (100 µg/mL) plus melatonin (1 µM) NCP suppressed AD-associated ROS and the intracellular Aβ production, aggregation, and accumulation in an ex vivo AD model based on cultured midbrain. The natural NCP showed high biocompatibility and demonstrated multimodal effectiveness for AD therapy [149].

Multifunctional NCP has been used to deliver melatonin more precisely to the breast cancer MCF-7 cells. In the study by Xie and colleagues, melatonin-loaded magnetic NC was generated by single emulsion using solvent extraction and evaporation; the mixture of 110 mg PLGA, 10 mg melatonin, and 30 mg iron oxide NPs was magnetically stirred until complete homogenization. Through alternating the magnetic field (e.g., magnetic heating), an effect of “inside-out” thermotherapy favored the sustainable release of melatonin from the PLGA, being the burst release attributable to the melatonin loaded near the NCP surface. High cellular uptake of the melatonin-incorporated NCP was confirmed, and the antiproliferative effect of melatonin on MCF-7 cells was dependent on concentration (ranging from 1 ng/mL to 1 μg/mL). Interestingly, nanothermal effect of heated melatonin showed a significant decrease in cell viability compared to non-heated melatonin. This nanoplatform highlights optimal activity of engineered melatonin-loaded multifunctional NCP [150].

To test the potential of magnetic NPs in myocardial hypertrophy (MH) and fibrosis, melatonin was incorporated into poly (lactide) polycarboxybetaine (PLGA-COOH) accompanied by cardiac homing peptide (CHP) and superparamagnetic iron oxide NPs (SPIONs) by emulsion method [151]. This CHP-mel@SPIONs nanoplatform demonstrated good EE of SPIONs (75.27 ± 3.1%) and melatonin (77.69 ± 6.04%), and no remnant magnetization or coercivity was observed. Using an animal model of pressure overload-induced MH, Zhao and colleagues reported that CHP-mel@SPIONs accumulated with more efficiency in the cardiac tissue compared with mel@SPIONs method; this was especially observed in the presence of an external magnetic field. The use of low doses ameliorated the MH as confirmed by echocardiography, histopathological features, and RT-PCR analysis.

Zinc oxide NPs (ZnO NPs) have unique physicochemical properties, low cost, and reduced environmental toxicity [155]. ZnO NPs are semiconductor materials (bandgap of ~3.3 eV) with important properties associated with thermal stability, robustness, and long-life compared with other metal oxide-derived materials (e.g., TiO_2_, WO_3_, SiO_2_, and Fe_2_O_3_). They can exist in hexagonal quartzite, cubic zincblende, and cubic rock salt, being the wurtzite phase the most stable structure. Due to the toxicity of ZnO NPs, melatonin has been associated to these NPs as a tissue protector [152]. The incubation of ZnO NPs with melatonin (100 µM) increased the activities of antioxidant enzymes in mouse brain. Administration of melatonin (10 mg/kg)-loaded ZnO NPs also protected against cyclophosphamide (CP)-induced reproductive damage, improving testicular and sperm parameters in CP-treated rats after 8 weeks of therapy [153].

A new drug delivery system based on multistimuli-responsive NPs has been generated based on the mitochondria structure for ischemic tissue repair. These biosmart NPs were designed with double shells as the two-layered membranes of mitochondria, the melatonin-loaded cores corresponded to the mitochondria matrix, and the circular DNA corresponded to mitochondrial DNA. Interestingly, melatonin-loaded cores simulated the protection mechanism synthesizing more melatonin in response to ischemia. At an acute ischemia, melatonin was faster released from mitochondria to scavenge ROS while activating its MT1 receptor to prevent cytochrome c release and apoptosis. This intelligent NP has been particularly studied to protect against myocardial ischemia [154]. Table 2 summarizes the main applications of the melatonin-associated metallic and non-metallic NCP.

## 11. Concluding Remarks and Perspectives

Nano drug delivery carriers have the ability to protect the incorporated agent from deterioration in physiological environments in a controlled manner while possessing long-lasting effects and lower side effects. Melatonin incorporated into different NCs displayed more accuracy and sustained delivery in multiple systems. For example, melatonin-loaded NPs showed superior antioxidant, anti-inflammatory, and antitumor properties in distinct cell types and biological tissues compared to that of melatonin in its free form.

On the basis of its therapeutic efficiency, melatonin-loaded lipid NCs showed antioxidant, anti-inflammatory, and antitumor effects in addition to favoring sleep, transdermal delivery, ocular absorption, protection against cell damage, and oocyte quality. Application of melatonin-loaded nanofibers and chitosan-based NPs resulted in skin and nerve regeneration, besides having antitumor effects. Furthermore, melatonin-loaded synthetic polymeric NCs protected against myocardial infarction, sepsis, ischemia-reperfusion-related damage, liver injury, brain and liver damage induced by ROS, and cancer, while stimulating bone engraftment, transdermal delivery, enthesis process, wound healing and cartilage regeneration. Also, metallic and non-metallic-based NCPs improved hepatocellular and testicular protection, and ameliorated fibrosis and heart hypertrophy; melatonin loaded into these NCPs further exhibited anticancer actions and reduced amyloid β protein aggregation related to the Alzheimer’s disease.

Since the type and size of the NCs can provoke some degrees of toxicity as well as delays in the elimination process, further long-term experimental studies should be performed to more appropriately define the specific and nonspecific distribution of the melatonin-containing NC and to test its safety in blood circulation and during renal excretion when considering prolonged periods of treatment. In addition, to identify the biocompatibility of melatonin with NPs, proper loading doses, route of administration, and interaction of these melatonin-loaded NPs with other cells (e.g., immune cells) or immune responses need a deeper investigation.

## Figures and Tables

**Figure 1 molecules-26-03562-f001:**
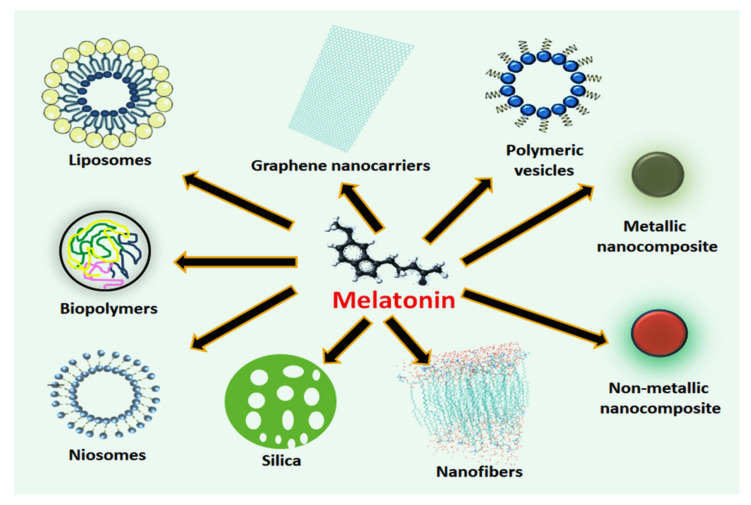
The use of melatonin-loaded nanostructures represents a promising therapeutic strategy. After being incorporated or adsorbed into solid lipidic, biopolymers or polymeric vesicles, silica NPs, nanofibers, graphene nanocarriers, and metallic or non-metallic NPs, melatonin is thought to achieve superior biological effects due to molecular protection and sustained release. These drug-delivery systems may be considered as an important approach to facilitate the permeability of the chemical agent while maintaining systemic safety during the treatment of complex pathologies and disease conditions.

**Figure 2 molecules-26-03562-f002:**
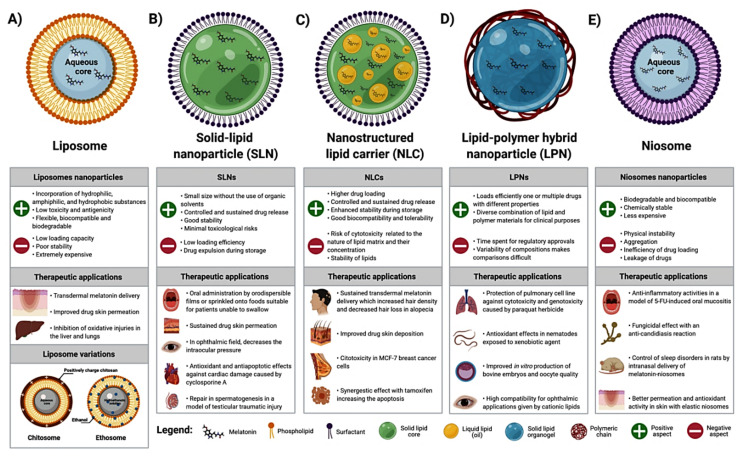
Structural representation of lipid nanosystems, main characteristics, and therapeutic applications of melatonin into liposomes (**A**), solid lipid NPs (SLN) (**B**), nanostructured lipid carriers (NLC) (**C**), lipid-polymer hybrid NPs (LPN) (**D**), and niosomes (**E**). Image was created using Biorender.com. Access in February 2021.

**Figure 3 molecules-26-03562-f003:**
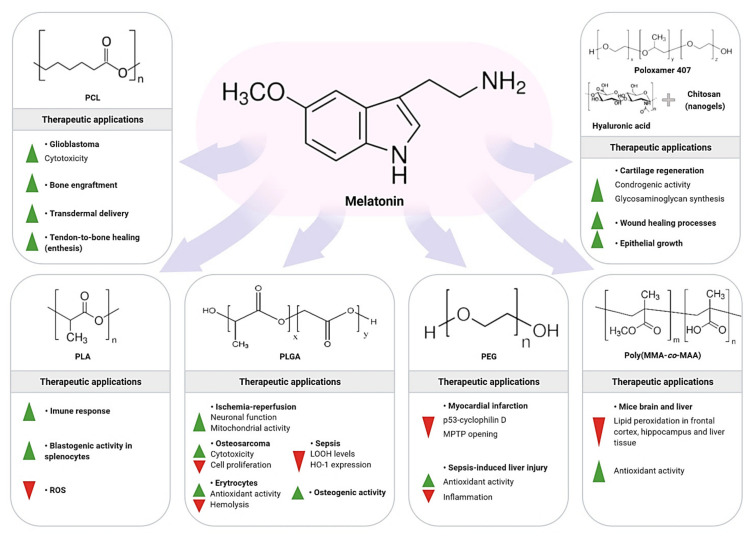
Melatonin incorporated into synthetic polymeric NPs or nanogel as NC provides a favorable biological response in different cells and damaged tissues. These activities are especially attributed to its antioxidant, anti-inflammatory, antitumor, and regenerative properties, in addition to the regulation of mitochondrial dynamics. PCL: polycaprolactone, PLA: poly-lactic acid, PLGA: poly (lactic-co-glycolic acid), PEG: polyethylene glycol, Poly (MMA-*co*-MAA): poly (methyl methacrylate-*co*-methacrylic acid). Green arrow = increased action, Red arrow = decreased action. Image was created using Biorender.com. Access in February 2021.

**Table 1 molecules-26-03562-t001:** Melatonin’s effects on different cell types and animal models after its incorporation into diverse NCs/nanoplatforms.

NC/Nanoplatform	Cell Type/Tissue/Nanomethod	Main Actions	Ref
Silica	HeLa cells	Longer polymers improved melatonin release and cell toxicity	[85]
Gaphene-dendrimeric system	Saos-2 and MG-63 cells	Antitumor action against osteosarcoma (higher cellular uptake and apoptosis)	[86]
Chitosan-polycaprolactone (PCL)/polyvinylalcohol (PVA)-melatonin	Rat skin	Improved skin regeneration, wound remodeling, and reduced inflammation	[87]
Bacterial cellulose nanofiber	Rat blood	Increased oral dissolution and bioavailability of melatonin	[88]
Ethylcellulose nanocapsules	Rabbit retinal ganglion cells	Slow in vitro release of melatonin and high corneal penetration	[89]
Cellulose acetate (CA), polyvinylpyrrolidinone (PV), and hydroxypropylmethylcellulose (HP)	Gastric-like fluids showing pH variations	Increased bioavailability of melatonin to treat sleep dysfunctions; control of the sleep-onset	[90,91]
Polyurethane (PU) and gelatin nanofibrils (GNFs)	Defected sciatic nerve of Wistar rats	Enhanced regenerative capacity of nerve and muscle function by melatonin	[92]
Lecithin/chitosan	U87MG and HepG2 cells	Improved melatonin release and permeability; reduced tumor growth and the genotoxic effect of drugs	[93,94,95,96]
Lecithin/chitosan	Lyophilization for storage	Prevented melatonin degradation and increased photostability	[56,97]
Lecithin/chitosan	Encapsulation efficiency in the nanoformulation	Promoted a slow release of melatonin	[95,96,98]
Chitosan buoyant microcapsules	Rats exposed to aflatoxin B1	Promoted superior antiapoptotic activity of melatonin	[99]

Abbreviations: NC: nanocarrier, HeLa: cervical cancer cell, Saos-2 and MG-63: osteosarcoma cell line, U87MG: human glioblastoma cell line, HepG2: hepatocellular carcinoma cells.

**Table 2 molecules-26-03562-t002:** Melatonin incorporated into metallic NPs and related metallic and non-metallic nanocomposites improved biological and biochemical-like processes in different animal models and cell types.

NC/Nanoplatform	Cell Type/Tissue/ Nanomethod	Main Actions	Ref
Gold NP	Murine macrophage cells	Improved cellular uptake	[143]
Gold NP	Rat testis tissue	Protected against testicular damage by reducing lipid peroxidation, TNF-α, and IL-1β level, and enhancing antioxidant capacity	[144]
Palladium NP	A549 cells	Increased lung cell toxicity via apoptosis and DNA oxidation, and reduced ATP content, and mitochondrial membrane potential	[145]
Selenium NP	Mouse model of liver injury	Improved hepatocellular protection by reducing the activity of aminotransferase, the extent of hepatic cell damage, and migration rate of inflammatory cells	[148]
Dopamine-melatonin nanocomposite	SH-SY5 cells; Balb/c mice	Suppressed ROS and intracellular Aβ production and aggregation in cultured midbrain cells of adult mice with Alzheimer’s disease	[149]
Magnetic nanocomposite	MCF-7 cells	Increased the antiproliferative effect of melatonin	[150]
Superparamagnetic iron oxide NP with PLGA-COOH	Animal model of myocardial hypertrophy	Low doses of melatonin ameliorated fibrosis and myocardial hypertrophy	[151]
Zinc oxide NP	Mice brain tissue	Increased the activities of antioxidant enzymes	[152]
Zinc oxide NP	Rat testis	Protected against cyclophosphamide-induced reproductive damage	[153]
Mitochondria-resembling NP	Multistimuli-responsive NP	Favored melatonin release after ischemia improving ROS scavenging and preventing apoptosis	[154]

Abbreviations: NC: nanocarrier, NP: nanoparticle, MCF-7: human breast cancer cell, A549: non-small lung carcinoma *cell*, SH-SY5: human neuroblastoma cell.

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
