# Peer review of "Melatonin-Loaded Nanocarriers: New Horizons for Therapeutic Applications"

_molecules, 2021, doi:10.3390/molecules26123562_

Round 1
Reviewer 1 Report
In this contribution by Chuffa and co-workers, the authors reviewed melatonin-loaded nanocarriers for therapeutic applications. It is a well written and comprehensive review. I recommend it for publication in Molecules after the following points are addressed.
- Line 88, what is MT?
- Line 98, the abbreviation for solid lipid nanoparticles (appear first time in the ms.) is SLNs or SLN. Remove ‘and’ before solid.
- Figure 1, it is not proper to put ‘chitosan’ and ‘polymer’ as one category.
- Figure 2, the readers can’t find the difference between liposome and noisome. Please add some notes.
- Stability issue is important for liposomes. Zwitterionic polymers are excellent candidates for stabilizing the liposomes. Several recent studies (doi.org/10.1021/acs.langmuir.9b00610; doi.org/10.1021/la302433a) related to this topic should be included.
- Line 566, ‘IC50’ to ‘IC50’.
- In section 8, there are both hydrophilic and hydrophobic polymers used for NPs as carriers of melatonin. What’s the principle to choose a polymer? Please add some discussion about this point.
Author Response
In this contribution by Chuffa and co-workers, the authors reviewed melatonin-loaded nanocarriers for therapeutic applications. It is a well written and comprehensive review. I recommend it for publication in Molecules after the following points are addressed.
Response: We really appreciate all of the points/issues raised by the reviewer in order to improve the quality and scientific value of the manuscript.
1) Line 88, what is MT?
Response: This abbreviation is referred to melatonin receptor. We add the full name of the melatonin receptor (please see highlighted text on page 2, line 88).
2) Line 98, the abbreviation for solid lipid nanoparticles (appear first time in the ms.) is SLNs or SLN. Remove ‘and’ before solid.
Response: Thank you for advising us. This was poorly stated and the abbreviation of solid lipid nanoparticles (SLN) has been corrected. We also deleted “and” before solid.
3) Figure 1, it is not proper to put ‘chitosan’ and ‘polymer’ as one category.
Response: As suggested, we separated chitosan and polymer in the legend.
4) Figure 2, the readers can’t find the difference between liposome and niosome. Please add some notes.
Response: Thank you for raising this issue. In fact, the main difference between these nanocarriers is that liposome is basically formed by phospholipids (represented by yellow color molecule in the legend) while niosome is a non-ionic surfactant-based vesicle (represented by a purple color molecule). So, the presence of a surfactant is the differential aspect of the nanoparticle, and it was represented by this figure.
5) Stability issue is important for liposomes. Zwitterionic polymers are excellent candidates for stabilizing the liposomes. Several recent studies (doi.org/10.1021/acs.langmuir.9b00610; doi.org/10.1021/la302433a) related to this topic should be included.
Response: We thank this reviewer by adding this important point regarding the liposome stabilization. As suggested, new references addressing the role of Zwitterionic polymers to make liposome highly hydrophilic and with neutral charge and antifouling properties were added to the topic of liposomes (please see highlighted text on page 5).
6) Line 566, ‘IC50’ to ‘IC50’.
Response: Thank you for this observation. As recommended, we changed IC50 to IC50.
7) In section 8, there are both hydrophilic and hydrophobic polymers used for NPs as carriers of melatonin. What’s the principle to choose a polymer? Please add some discussion about this point.
Response: Thank you for raising this important issue regarding the different types of polymers. As requested, we added a short discussion on hydrophilic and hydrophobic polymers as potential carriers of melatonin before describing each one (please see highlighted text on page 13).
Reviewer 2 Report
The review entitled “Melatonin-loaded nanocarriers: new horizons for therapeutic applications” offers a nice and well-written overview about the use of nanocarriers to enhance the performance of melatonin.
Overall, the literature selection and the extensive number of examples well support the stay of the art and the needed to employ more extensively the use of nano-systhem to deliver melatonin.
Particularly appreciated were the deep explanation of the Melatonin encapsulation into lipid-based nanosystems in a relatively easy way, so that even a “non-expert” reader can understand the importance of the lipid particles to improve drug solubility and permeability.
However, from my point of view a section where all the already available melatonin based-formulations on the market are briefly described is missing. What is also missing is a clear description of the different administration routes. I would add a paragraph in which the readers are driven through the available administration routes and I would list the pro and cons of each.

Author Response
The review entitled “Melatonin-loaded nanocarriers: new horizons for therapeutic applications” offers a nice and well-written overview about the use of nanocarriers to enhance the performance of melatonin.
Overall, the literature selection and the extensive number of examples well support the stay of the art and the needed to employ more extensively the use of nano-systhem to deliver melatonin.
Particularly appreciated were the deep explanation of the Melatonin encapsulation into lipid-based nanosystems in a relatively easy way, so that even a “non-expert” reader can understand the importance of the lipid particles to improve drug solubility and permeability.
Response: We really appreciate all of the points/issues raised by the reviewer in order to improve the quality and scientific value of the manuscript. We did our best to address all raised issues.
However, from my point of view a section where all the already available melatonin based-formulations on the market are briefly described is missing. What is also missing is a clear description of the different administration routes. I would add a paragraph in which the readers are driven through the available administration routes and I would list the pro and cons of each.
Response: Thanks for raising this important issue. As suggested, we improved this section by adding new information on melatonin-based formulations. Also, a new paragraph reinforcing the available administration routes of melatonin mentioning the pro and cons of each individual route were provided (please see highlighted text on page 3).
Reviewer 3 Report
Manuscript Number: molecules-1243335
Manuscript Title: “Melatonin-loaded nanocarriers: new horizons for therapeutic
Applications” by
This review is aimed at summarizing the progress of nano-scale delivery systems for melatonin. These nano-scale delivery platforms or nano-carriers include lipid-based vesicles, polymeric vesicles, non-ionic surfactant-based vesicles, charge carriers in graphene, electro spun nanofibers, silica-based carriers, metallic and non-metallic nanocomposites. The authors made considerable efforts to collect and summarize the relevant research work from various aspects (biological and material; in vitro and in vivo).
Overall, this review is written in a fairly straightforward manner with reasonable logic and plenty of useful information. In addition, inclusion of 3 schematic diagrams/3figures and 2 tables can definitely help readers to capture the key information/concepts of the findings in the related research fields. This review also cites some of the latest publications. In my opinion, this review article can be accepted in its current state.
Author Response
This review is aimed at summarizing the progress of nano-scale delivery systems for melatonin. These nano-scale delivery platforms or nano-carriers include lipid-based vesicles, polymeric vesicles, non-ionic surfactant-based vesicles, charge carriers in graphene, electro spun nanofibers, silica-based carriers, metallic and non-metallic nanocomposites. The authors made considerable efforts to collect and summarize the relevant research work from various aspects (biological and material; in vitro and in vivo).
Overall, this review is written in a fairly straightforward manner with reasonable logic and plenty of useful information. In addition, inclusion of 3 schematic diagrams/3figures and 2 tables can definitely help readers to capture the key information/concepts of the findings in the related research fields. This review also cites some of the latest publications. In my opinion, this review article can be accepted in its current state.
Response: We really appreciate all the comments and recognition of our review paper. We are very glad that our manuscript met the reviewer’s expectations.
Round 2
Reviewer 1 Report
For the comment 3 in my last report, it is still not well addressed. In the figure 1, they should change the 'polymer' into 'polymeric vesicle' and 'chitosan' to 'biopolymer'. Chitosan is one typical polymer or biopolymer, but it can't be listed as a category.
After this small point, I will recommend it for publication.